# A Review of Zoonotic Babesiosis as an Emerging Public Health Threat in Asia

**DOI:** 10.3390/pathogens11010023

**Published:** 2021-12-24

**Authors:** Sabir Hussain, Abrar Hussain, Muhammad Umair Aziz, Baolin Song, Jehan Zeb, David George, Jun Li, Olivier Sparagano

**Affiliations:** 1Department of Infectious Diseases and Public Health, Jockey Club College of Veterinary Medicine and Life Sciences, City University of Hong Kong, Kowloon, Hong Kong SAR 999077, China; muhamaziz3-c@my.cityu.edu.hk (M.U.A.); Baolin.Song@my.cityu.edu.hk (B.S.); jehanzeb2@cityu.edu.hk (J.Z.); jun.li@cityu.edu.hk (J.L.); 2Department of Epidemiology and Public Health, University of Veterinary and Animal Sciences, Lahore 54600, Pakistan; 2018-mphil-2093@uvas.edu.pk; 3School of Natural and Environmental Sciences, Newcastle University, Newcastle upon Tyne NE1 7RU, UK; david.george1@newcastle.ac.uk

**Keywords:** babesiosis, zoonosis, humans, animals, transmission, Asia

## Abstract

Zoonotic babesiosis poses a serious health risk in many parts of the world. Its emergence in Asia is thus a cause for significant concern, demanding that appropriate control measures are implemented to suppress its spread in this region. This study focuses on zoonotic *Babesia* species reported in Asia, offering an extensive review of those species reported in animals and humans. We reported 11 studies finding zoonotic *Babesia* species in animals and 16 in humans. In China, the most prevalent species was found to be *Babesia microti*, reported in both humans (n = 10) and wild and domesticated animals (n = 4). In Korea, only two studies reported human babesiosis, with a further two studies reporting *Babesia microti* in wild animals. *Babesia microti* was also reported in wild animal populations in Thailand and Japan, with evidence of human case reports also found in Singapore, Mongolia and India. This is the first review to report zoonotic babesiosis in humans and animals in Asia, highlighting concerns for future public health in this region. Further investigations of zoonotic species of *Babesia* in animal populations are required to confirm the actual zoonotic threat of babesiosis in Asia, as well as its possible transmission routes.

## 1. Introduction

Babesiosis is a vector-borne disease of profound public health significance. Three *Babesia* species are primarily responsible for causing zoonotic babesiosis in many parts of the world, these being *B. divergens*, *B. microti* and *B. venatorum* [1]. *Babesia divergens* originated from bovines and has been reported in cases of human babesiosis in Europe [2], whereas *B. microti* predominantly spreads from rodents in North America and Asia [3]. *Babesia venatorum* is present in other parts of the world, often originating from Europe [3]. Ticks from the genus *Ixodes* are the main vectors for babesiosis, where their geographical distribution determines the prevalence of the pathogens involved [4]. Worryingly, the geographical range of *Ixodes* ticks is expanding due to climate change, where, as a result, it is projected that incidences of Lyme disease will increase by 20% in the next decade [5]. Given that *Ixodes* ticks also vector *Babesia microti* and *Borrelia burgdorferi*, we can probably expect similar increases in the incidence of *Babesia* spp.

Across Europe, North America and Asia, the main vector species of *Ixodes* are *I. ricinus, I. scapularis,* and *I. persulcatus*, respectively [2,6]. Babesiosis is acquired by humans through infected tick bites [7], though it can also be spread via blood transfusion [8] and through transplacental transfer from mothers [9]. Babesiosis symptoms in humans include headache, fatigue, loss of appetite, fever, chills, nausea, and shortness of breath, with the elderly at higher risk of severe symptoms such as hepatomegaly, kidney failure, hemolytic anaemia, splenomegaly and splenic complications, potentially resulting in death [10]. Interestingly, unlike other severe complications, the splenic complication of acute babesiosis is not associated with the increase in parasitemia or related to the host immune status. Blood smear examination is commonly practiced for diagnosis, and a combination of atovaquone and azithromycin, or a course of clindamycin and quinine, is often used to treat mild and severe cases, respectively [6,11].

In wild and domesticated animals, the severity of babesiosis depends upon a range of factors. In the case of *B*. *divergens* infection in cattle, severity is determined by the immune status of the animal, infective dose, virulence of the strain, and intensity of tick infestation [12]. Many mammals, and a few bird species, are known hosts of zoonotic *Babesia* species, as well as the ticks that vector them during their complex life cycle, transmitting these pathogens either transovarially or transstadially. Cattle, roe deer, and other domestic and wild ruminants are reservoirs for *B*. *divergens* and *B. venatorum,* while meadow voles, white-footed mice, cottontail rabbits, and other small mammals are reservoirs of *B. microti* [13].

As a region, Asia is currently experiencing extensive human modification of natural habitats, placing its biodiversity at risk [14]. The damage to Asia’s habitats is also increasing the risk of zoonotic disease spread, benefitting known reservoirs of zoonotic diseases, especially rodents (which are more adaptable and resistant to habitat change) [15]. Zoonotic transmission of babesiosis in Asia is consequently increasing, the impact of which is being further exacerbated by a lack of molecular diagnostic techniques and clinical diagnostic expertise, insufficient medical awareness and low throughput capacity to detect the pathogens responsible [16]. Due to its low incidence, human babesiosis is an under-represented disease in Asia, and its occurrence has not been properly investigated in this region. Nevertheless, it is known that distinct *Babesia* species are responsible for the spread of babesiosis across Asia. *Babesia microti*, for example, appears primarily responsible for disease spread in Japan [17], *Babesia* sp. KO1 in Korea [18] and *B. microti*-like in both Taiwan [19] and Mainland China, where *B. venatorum* and *Babesia* sp. XXB/HangZhou have also been reported as causal agents [20]. It was also reported that rodents and their associated ticks act as major reservoirs of *Babesia* spp. in Asia, including for *B. microti and B. microti-*like [21]. However, key knowledge gaps remain regarding our understanding of the role of various transmission factors in this region, including that of alternative disease reservoirs and the ecological conditions that facilitate zoonotic babesiosis spread. Information on the actual status of zoonotic babesiosis in Asia, in terms of its species distribution, reservoir diversity, intensity and prevalence, is lacking. 

With the above in mind, the current study aimed to increase our understanding of zoonotic babesiosis as an emerging public health threat in Asia, focusing on the zoonotic dimension of babesiosis in multiple Asian countries by exploring human case reports and future perspectives of this disease.

## 2. Materials and Methods

The literature regarding babesiosis was interrogated using three separate databases (Scopus, Google Scholar and PubMed) and a series of pre-determined search terms/keywords: “babesiosis”, “human babesiosis”, “*Babesia*”, “zoonotic”, “zoonoses”, and “Asia”. We compiled all studies which included the prevalence of zoonotic species of *Babesia* in animals and also examined case reports of babesiosis in humans, accessing the literature published prior to 15 October 2021. Studies that were not available in full online were retrieved via the “Library Find” database at the City University of Hong Kong. 

A total of 15 studies relating to zoonotic babesiosis in animals were retrieved from our literature search, with four studies subsequently removed to avoid duplication. Seventeen studies were returned relating to human babesiosis, with three of these later excluded for either the same reason or due to an unconfirmed diagnosis being presented. From the resulting final list of studies, two separate summary tables were compiled; one for animals and one for human-related reports. In these tables, animal-related studies were presented according to the year of study, country, method of disease detection, host species, *Babesia* species, total samples, prevalence and 95% CI, while human-related studies were organised according to their country/location, patient details, patient history, diagnostic technique used, and species diagnosed. Maps of reported cases were generated and georeferenced using Microsoft Excel, version 2019.

## 3. Results and Discussion

### 3.1. China

In China, zoonotic babesiosis was detected in four studies, from rodents and small mammals, with an average prevalence of 6.26% and all studies confirming *B. microti* in samples through PCR and sequencing (Table 1) (Figure 1). For human babesiosis, 181 cases were reported from 10 different studies across numerous Chinese regions, with 23.8% (n = 43) of cases being “suspected” and 76.2% (n = 138) being “confirmed”. Multiple zoonotic species of *Babesia*, including *B. divergens, B. microti, B. venatorum, B.* sp. XXB/Hangzhou, and *B. crassa-*like, were detected in the human population through various techniques, including microscopy, IFA (immunofluorescence assay), FISH (fluorescence in situ hybridisation), and PCR. Based on a comprehensive review of each case report, in 50% (n = 5) of studies, tick bites were noted as a possible route of transmission, based on consideration of patient histories. Tick bites were confirmed as the transmission route in a further 10% (n = 1) of cases. In 20% (n = 2) of studies, blood transfusion was determined as having caused disease spread, while remaining studies were unable to identify a possible route of transmission (Table 2) (Figure 2). 

During the last two decades, more than 100 patients from various provinces in China have tested positive for *B. microti,* making this pathogen the dominant cause of babesiosis in the Chinese (human) population. *Babesia microti* was reported in the Chinese population for the first time in 2011, in Zhejiang province, as confirmed using PCR [45]. *Babesia crassa*-like also contributes to human babesiosis in China, having been first reported as doing so in 2015. It has since been confirmed as a novel *Babesia* species in Heilongjiang, where between 2015 and 2016, it caused 58 cases of babesiosis, from a sample size of 1125 participants with tick bite histories [44]. *Babesia venatorum* is also circulating in the human population and causing infection in China. According to one study undertaken in the province of Heilongjiang, 48 positive cases of this species were reported from a sample of 2912 individuals with a history of tick bites [41]. Another species, *B. divergens*, appears less commonly associated with Chinese human babesiosis infection, only being reported in two cases from Shandong province, as confirmed through PCR amplification and sequencing [39]. Another novel species of *Babesia* causing human babesiosis, *Babesia* sp. XXB/Hangzhou was also reported in China in 2015, albeit as a single case in a 42-year-old male in Hangzhou, Zhejiang province [42].

According to the current review, most of the babesiosis patients identified presented with a history of tick bites or blood transfusion, both of which are major factors for increased risk of transmission. Areas with abundant vegetation are also associated with increased risk of human babesiosis cases as a probable result of their potential to harbour vector populations. Dense vegetation is present in the southern areas of Zhejiang and Guangxi, while hilly forested areas predominate in Heilongjiang in northeastern China [20,41]. Both habitats provide suitable environments for ticks and their hosts, such as rodents and small mammals, and the studies which reported zoonotic *Babesia* species revealed that rodents and small mammals were major sources for enzootic cycling and transmission of *B. microti* in China. Many rodent and mammalian species are involved in the enzootic maintenance and transmission of *B. microti*, such as *Niviventer* spp., *Macaca* spp., *Rattus* spp., and *Citellus* spp., and all are capable of harbouring this zoonotic pathogen [35]. These mammals contribute to the widespread distribution of ticks carrying zoonotic species of *Babesia*, and people entering habitats where these animals are abundant (e.g., forest) should adopt adequate protection measures to limit their exposure to ticks and pathogens they may carry. 

### 3.2. Korea

The review of zoonotic babesiosis in Korea found that small mammals and wild rescued mammals from the region had tested positive for *B. microti*, confirmed through PCR in two studies with a prevalence of 2.1% and 5.7%, respectively (Table 1) (Figure 1). It also demonstrated that only three case reports of human babesiosis had been reported to date in Korea. One of these was from a 75-year-old female in Gurae, Jeon-Nam province, resulting from *Babesia sensusstricto* as detected through PCR; patient history revealed that the afflicted individual had received blood transfusions twice, where, on the basis of her symptoms, she was initially diagnosed with malaria (later confirmed as babesiosis when antimalarial drugs were found ineffective). In the second case report, two cases attributed to *B. motasi* reported from Hoengseong-gun and Gangwon-do, with patients having a history of dizziness and weakness, while the third case report provides the evidence for two cases of human babesiosis in females with a history of travelling, and both cases were *Babesia microti* confirmed through PCR (Table 2) (Figure 2).

There are many animal species that can be affected by *B.*
*microti*, but studies in Korea regarding the reservoir competence of many wildlife hosts are limited. A few studies included in our literature review reported the first detection of *B. microti* infection in Chinese water deer and Eurasian badgers, along with other small wild mammals. *Babesia microti* has been reported in other mammalian species in the US, including foxes [46], short-tailed shrews [47], raccoons [48], eastern cottontail rabbits and eastern chipmunks [49]. In Europe, small mammals play a very important role in spreading *B. microti*, with 16 species of insectivores and rodents documented as hosting this pathogen [50]. Small mammals are thus widely reported reservoirs of *B. microti* and play a major role in maintaining this pathogen in nature, with ticks feeding upon these hosts to drive the transmission cycle [47]. There is certainly opportunity for *B. microti* to circulate in the diverse small mammal populations of Korea, but its occurrence and prevalence within these species is still largely unknown due to a limited number of studies on the topic. 

### 3.3. Japan

In Japan, four studies have detected zoonotic species of *Babesia* from animals to date, confirming results through PCR. Two studies reported *B. microti* from wild rodents, with prevalence of 13.4% and 45.2%, while another study reported *B. microti* from field rodents with a prevalence of 14.6%. A further important zoonotic *Babesia* species, *B. divergens*, was reported from wild sika deer in Japan, with a prevalence of 6.6% (Table 1, Figure 1 and Figure 2). Only one PCR-confirmed human babesiosis case was reported in Japan, in a 40-year-old male from Kobe, Hyogo Prefecture. The patient was admitted to the hospital due to gastric disorder and bleeding, where, as a result, he received circa 2 liters of blood via transfusion. After one month, the gastric ulcer was cured, but the patient presented with dark-coloured urine and anaemia. *Babesia*-like intraerythrocytic parasites were subsequently found through a Giemsa-stained blood smear, and *B. microti* was later confirmed through IFA and PCR [51].

*Babesia microti* is correlated with the distribution of rodents in Japan [52]. In the studies reviewed here, sampling areas were not distinguished on the basis of their suitability as rodent habitats nor other environmental factors, but *B. microti* parasites have been isolated from wild rodents in a variety of areas [17]. The results from a study in Central Croatia indicated that two species of wild rodents, *Apodemus flavicollis* and *Myodes glareolus*, were commonly infected with *B. microti* [53]. *Babesia microti* has also been detected in European rodents, including in several areas of Poland [54]. Another study indicated that sika deer commonly tested positive for *B. divergens* [32]. This suggests an urgent need for further research to investigate the risk of this pathogen being transmitted from these deer (which are common in Japan) to humans.

### 3.4. Singapore

The first case of human babesiosis in Singapore was reported in 2018 in a 37-year-old man. The National Public Health Laboratory in Singapore confirmed the cause as being *B. microti* by performing microscopy and PCR. This remains the only zoonotic case reported in humans to date, where the afflicted individual was believed to have acquired the infection after being bitten by a tick in the US, highlighting the presence of ticks carrying zoonotic babesiosis in America (Table 2) Figure 1. 

This single case of zoonotic babesiosis in Singapore revealed that any traveler having a history of tick bites and relevant symptoms should be adequately investigated, especially when returning from countries where tick-borne diseases are endemic. In the US, *Babesia* is most commonly transmitted through blood transfusion, so all travellers must be screened as per Food and Drug Administrations and recommendations [55]. *Babesia microti* is rarely found in areas other than the US, but babesiosis due to *B. microti* has been frequently reported in travellers returning from North America [56]. A similar case report described *B. microti* presence in an 82-year-old man who travelled to the US and returned to France with severe fever and hemophagocytosis [57]. 

Although cases of zoonotic babesiosis are currently extremely rare in Singapore, the potential for this disease to circulate in the region exists, at least in domesticated animals. A preliminary survey of tick fauna in this country indicated that various tick species in Singapore could serve as vectors for bovine and canine *babesia* transmission, though, as yet, no vector for human babesiosis has been reported [58].

### 3.5. Other Countries

*Babesia microti* has been reported in Cambodia, Laos and Thailand, identified from wild rodents with a prevalence of 5.3%. In the Selenge province of Mongolia, 100 asymptomatic farmers were screened, and 7% were found to be positive for *B. microti* antibody, with 3% having amplifiable *B. microti* DNA in their blood (Table 1 and Figure 1). Human babesiosis was also reported in India, though only in a single case of a man in the area of Baroda (Gujarat), where the genus *Babesia* was identified through microscopy (Table 2 and Figure 2). Here, a 51-year-old male presented in a private hospital with fever, loss of appetite and vomiting, with *Babesia* morphologically identified from smears taken after antibiotics failed to treat the patients’ symptoms. Antigen testing for Plasmodia was negative, which showed that the patient was not affected by malaria [35,59]. As the patient did not have any exposure to tick bites, nor any history of visiting *Babesia* endemic areas, an unknown indigenous source of infection was assumed, which in itself warrants further investigation.

## 4. Conclusions

To our knowledge, this is the first literature review compiling reports of zoonotic babesiosis from animal and human cases in Asia. After extensive searching, we found that the highest number of zoonotic species of babesiosis in animals and human case reports have emanated from China, followed by Korea, where the most common zoonotic species circulating in Asia appear to be *B. microti* and *B. divergens.* The most common zoonotic hosts identified were rodents and other wild mammals. The zoonotic catalogue presented here will help in identifying potential hotspot areas in Asia where these zoonotic species are circulating, informing control strategies to prevent disease transmission.

## Figures and Tables

**Figure 1 pathogens-11-00023-f001:**
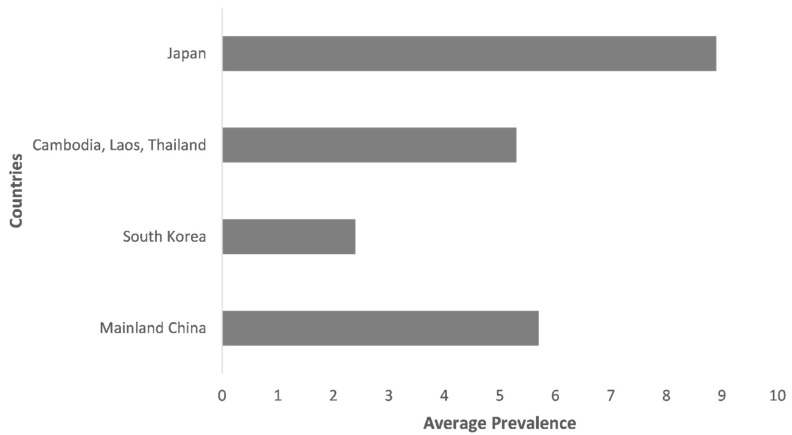
Prevalence of zoonotic babesiosis in animal populations of Asia.

**Figure 2 pathogens-11-00023-f002:**
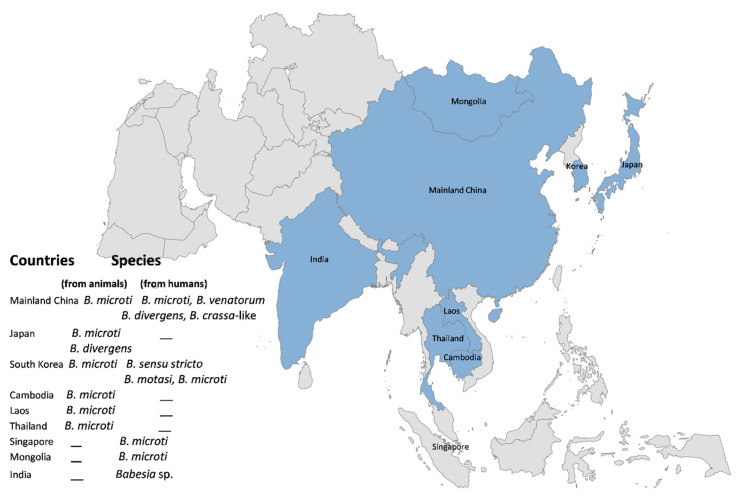
Countries with babesiosis in animal and human populations of Asia.

**Table 1 pathogens-11-00023-t001:** Zoonotic species of *Babesia* reported in animal populations of Asia.

Year of Study	Country and Region	Continent	Method of Identification	Sample Site/Host	Species	Sample Size	Positive	Prevalence %	CI 95%	Reference
2009–2011	China	Asia	PCR/Seq	Wild rodents	*B. microti*	1672	72	4.3	3.4–5.4	[22]
2009–2011	China	Asia	PCR/Seq	Small mammals	*B. microti*	2204	53	2.4	1.8–3.1	[23]
2018	China	Asia	PCR/Seq	Small mammals	*B. microti*	1391	168	12.1	10.4–13.9	[24]
2002–2005	China and Taiwan	Asia	PCR/Seq	Field rodents	*B. microti*	68	15	22.1	12.9–33.8	[25]
2008	Korea	Asia	PCR/Seq	Small mammals	*B. microti*	667	14	2.1	1.2–3.5	[26]
2008–2009	Korea	Asia	PCR/Seq	Rescued wild animals	*B. microti*	70	4	5.7	1.6–14.0	[27]
2008–2009	Cambodia, Laos, Thailand	Asia	Nested PCR	Wild rodents	*B. microti*	1439	76	5.3	4.2–6.6	[28]
1998	Japan	Asia	PCR/Seq	Wild rodents	*B. microti*	97	13	13.4	7.3–21.8	[29]
2003–2005	Japan	Asia	PCR/Seq	Field rodents	*B. microti*	247	36	14.6	10.4–19.6	[30]
2000–2004	Japan	Asia	PCR/Seq	Wild rodents	*B. microti*	62	28	45.2	32.5–58.3	[31]
2012–2018	Japan	Asia	PCR	Wild Sika deer	*B. divergens*	1747	116	6.6	5.5–7.9	[32]

**Table 2 pathogens-11-00023-t002:** Cases of human babesiosis reported in Asia.

Year	Species	Geographical Location	Country	Number of Cases	Diagnostic Technique	Potential Transmission Route	Gender	Age Range (Years)	Reference
2006	*Babesia sensu stricto*	Gurae, Jeon-nam	Korea	1	Microscopy, PCR	Blood transfusion	Female	75	[18,33]
2005	*B. motasi*	Hoengseong-gun, Gangwon-do	Korea	2	Microscopy, PCR	tick bite	Male	70	[32]
2018	*B. microti*	Incheon	Korea	2	PCR	–	Female	50–72	[34]
2004	*Babesia* sp.	Baroda (Gujarat)	India	1	Microscopy	–	Male	51	[35]
2011	*B. microti*	Selenge	Mongolia	3	IFA, PCR	–	Male	–	[36]
1999	*B. microti*	Kobe, Hyogo Prefecture	Japan	1	IFA, PCR	Blood transfusion	Male	40	[37]
2018	*B. microti*	Tan Tock Seng Hospital	Singapore	1	Microscopy, PCR	Tick bites	Male	37	[16]
1994	*B. microti*-like	Southern Taiwan	China	2	Microscopy, IFA, inoculation	–	Female	51, −	[19]
2000	*Babesia* sp.	Hangzhou, Zhejiang	China	1	Microscopy	Blood transfusion	Male	36	[38]
2009	*B. divergens*	Tai’an, Shandong	China	2	PCR	–	Male	–	[39]
2010	*B. microti*-like	Yunnan	China	1	Microscopy, IFA	Tick bites	Female	46	[40]
2012–2013	*B. microti*	Tengchong, Yunnan	China	10	Microscopy, PCR	Blood transfusion, tick bites	6 males	22–45	[20]
2012	*B. venatorum*	Pishan, Xinjiang	China	1	Microscopy, PCR, inoculation	Tick bites	Male	8	[40]
2011–2014	*B. venatorum*	Heilongjiang	China	48 ^a^	PCR, microscopy, FISH, inoculation	Tick bites	–	0.6–75	[41]
2015	*B*. sp. XXB/Hangzhou	Hangzhou, Zhejiang	China	1	Microscopy, PCR	–	Male	42	[42]
2013–2015	*B. microti*	Guangxi Zhuang	China	48	Microscopy, PCR, IFA	–	–	–	[43]
2015–2016	*B. crassa*-like	Heilongjiang	China	58 ^b^	Microscopy, PCR	Tick bites	19 males	4–72	[44]

a. Sixteen were suspected cases; b. Twenty-seven were suspected cases.

## Data Availability

The dataset generated for this study is available from the corresponding author upon request.

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
