# Peer review of "A Review of Zoonotic Babesiosis as an Emerging Public Health Threat in Asia"

_pathogens, 2021, doi:10.3390/pathogens11010023_

Round 1

Reviewer 1 Report

This is an excellent review of this issue in Asia and provide an excellent ground base to extend further work.

Author Response

Thank you

Enclosed here file for your kind consideration

Reviewer 2 Report

In the current review authors describe Babesia spp in Asia, both in animals and in humans. Given the climate change, expansion of Ixodes ticks, babesiosis is here to stay, and furthermore increase in incidence.

I find the manuscript important, and it does contribute to the current knowledge on this topic. It is well written and easy to understand. 

This being said, I do have the following comments that I would like authors to address before manuscript is re-considered for publication

  1. Line 36 in introduction. It can be added that due to climate change, Ixodes tick is expanding its geographical presence as documented in recent study ( https://pubmed.ncbi.nlm.nih.gov/30473737/. 
  2. In above study it is projected that Lyme disease will increase for 20% in the next decade.  Given that the vector Ixodes tick is the same for Babesia microti and Borrelia burgdorferi, we can probably expect the similar changes and increase in the incidence of Babesia spp.
  3. Line 42 in introduction. Splenic complication of Babesia microti in humans have been recently reviewed. It should be added that patients also suffer from splenic infarct and rupture in addition to splenomealy that you mentioned. Interestingly, unlike other severe complication , splenic complication of acute babesiosis do NOT correlate with burden of parasitemia or host immune status ( usually healthy males with low parasitemia suffer from these)
  4. Methodology should be done following PRISMA guidelines for systematic review
  5. Results- there is a discrepancy in reporting: Line 96-- 43 suspected + 129 confirmed cases is 172, BUT when you calculated cases in table 1 and table to it is 175; Please correct this discrepancy or provide explanation for the same
  6. For each human case in different countries ( India, Korea, Singapore) authors should report if the case was imported or autohton and provide the reference in the text
  7. Line 211- was malaria ruled out here? microscopy is not ideal diagnostic method to differentiate the two, as they are frequently similar so inexperienced eye can make a mistake easily

Author Response

Thank you

Enclosed here file that contains all answers.

Reviewer 3 Report

Reviewer’s comments to Hussain et al.,

This manuscript summarized the zoonotic babesiosis in human and animals in Asia. The human babesiosis is important topic in parasitology field and researchers in Asia should keep in mind about this disease. The concept of this manuscript is fine, however in my opinion, it requires major revision before ready for publication. My comments are shown below.

Major comments

  1. The style of the article

This manuscript is very confusing either review paper or meta-analysis original paper. In my opinion, review paper is describing overview of the field and not limiting information source from specific database search. I think the authors should revise following points.

I. The authors should include all available confirmed cases of babesiosis. I read another review pathogens10111447 (Kumar et al., 2021) and there were several human babesiosis report in Asian countries.

II. Page 1 line 22, Page 7 line 218: Since this is a review paper (based on already reported data), no need to write “first study”.

III. Page 2 line 74, 91: As far as I know, I never see “Material and Method” and “Result and Discussion” in review papers. Please change the paragraph structure.

  1. The definition of zoonotic Babesia species

The authors should clarify the zoonotic Babesia species. I think B. microti and B. divergens are widely accepted as zoonotic species, but historically there were some B. bovis cases in human. I think the authors may not intend to include the bovine B. bovis data in this review, and the authors should clearly explain the Babesia species they will discuss in this review.

Minor comments

  1. Page 1 line 12: The statement “Zoonotic babesiosis is expanding its borders”, is it based on any data?
  2. Page 2 line 52: B. (half space) divergens.
  3. Table 1, left side of [47]: 21.8
  4. Figure 1 and 2: The figures are less importance and even misleading. I recommend to delete them (or extensive revision is necessary). I. The data is unclear and I think the authors have miscalculated. Regarding Fig 1 Japan, when I calculated 4 data from Table1, result will be 9.0ï¼… (193/2153). Fig2 China, when I added 8 data, result will be 64, however I’m wondering the data of Heilongjiang. The second data 58, 27 were suspected cases but 31 are confirmed case isn’t it? If so, the data will be 95. II. The style of the figures is different between a and 2, and the style of Fig 2 looks like dot plot.
  5. Figure 3: India; the parasite was detected from human. Babesia may Babesia sp.
  6. Page 6 line 151: sensu (half space) stricto. In addition, are there any specific reason that the authors explain it as Babesia sense stricto but not Babesia sp.?
  7. Reference: The species name should be italicized.

Author Response

Thank you

Enclosed here file that contains all answers given point by point

Round 2

Reviewer 2 Report

Line 49- reference is missing- https://www.ncbi.nlm.nih.gov/pmc/articles/PMC7275217/

I still believe that using PRISMA guidelines the manuscript would gain on the strength. However, this decision will be up to the editor to insist on or not. The reason why I am stating this is that by using the PRISMA you minimize the risk of omitting some of the articles. For example, the following paper was missed https://pubmed.ncbi.nlm.nih.gov/30630283/

Author Response

The manuscript has been revised by native English speaker and all spell check and grammatical mistakes have been corrected in the revised manuscript.

In the line 49, reference has been cited in the revised manuscript.

This is general review paper so we did not follow inclusion and exclusion criteria as per PRISMA guidelines. But through rigorous literature search we included all the research articles of Asia reporting the zoonotic babesiosis. The suggested article has been added and cited in the manuscript.

Thank you

Reviewer 3 Report

The authors have revised manuscript and I have just one comment .

Point4. Figure 1: I disagree with the authors’ calculation method. Since the authors know the sample numbers, they should calculate by positive sample number/total sample number. (Or are there any clear reason to neglect sample size difference?)

Author Response

The manuscript has been revised by native English speaker and all spell check and grammatical mistakes have been corrected in the revised manuscript.

Figure 1 has been updated according to the suggested method by using the positive sample number/total sample number.

Thank you